# #INSTAG: INSTRUCTION TAGGING FOR ANALYZING SUPERVISED FINE-TUNING OF LARGE LANGUAGE MODELS

**Keming Lu**[\*] **& Hongyi Yuan**[\*][†]**& Zheng Yuan & Runji Lin**[†]
Alibaba DAMO Academy
{lukeming.lkm,yuanhongyi.yhy,yuanzheng.yuanzhen,linrunji.lrj}@alibaba-inc.com

**Junyang Lin & Chuanqi Tan & Chang Zhou & Jingren Zhou**
Alibaba DAMO Academy
{junyang.ljy,chuanqi.tcq,ericzhou.zc,jingren.zhou}@alibaba-inc.com

## ABSTRACT

Pre-trained large language models (LLMs) can understand and align with human instructions by supervised fine-tuning (SFT). It is commonly believed that diverse and complex SFT data are of the essence to enable good instruction-following abilities. However, such diversity and complexity are obscure and lack quantitative analyses. In this work, we propose INSTAG, an open-set instruction tagging method, to identify semantics and intentions of human instructions by tags that provide access to definitions and quantified analyses of instruction diversity and complexity. We obtain 6.6K fine-grained tags to describe instructions from popular open-sourced SFT datasets comprehensively. We find that the abilities of aligned LLMs benefit from more diverse and complex instructions in SFT data. Based on this observation, we propose a data sampling procedure based on INSTAG, and select 6K diverse and complex samples from open-source datasets for SFT. The resulting models, TAGLM, outperform open-source models based on considerably larger SFT data evaluated by MT-Bench, echoing the importance of instruction diversity and complexity and the effectiveness of INSTAG. INSTAG has robust potential to be extended to more applications beyond the data selection as it provides an effective way to analyze the distribution of instructions.

## 1 INTRODUCTION

The contemporary chatbots, such as GPT-4 (OpenAI, 2023), have brought to the forefront of artificial generative intelligence with their superior and versatile abilities in real-world task solving. Such abilities are unlocked by fine-tuning pre-trained large language models (LLMs) to align human preference, and well-aligned LLMs can precisely recognize human intentions and properly formalize responses expressed in natural languages. There have been proposed various techniques to achieve such alignment of enabling pre-trained models to comprehend and execute diverse instructions effectively, including supervised fine-tuning (SFT) (Taori et al., 2023; Chiang et al., 2023), rejection sampling (Yuan et al., 2023b; Song et al., 2023; Rafailov et al., 2023), and reinforcement learning with human feedback (RLHF) (Bai et al., 2022a; Ouyang et al., 2022; Touvron et al., 2023b).

Especially, SFT for alignment is widely studied by recent research, which is generally formalized in a multi-turn utterance manner, and each turn is composed of a human query and a corresponding response well-aligned with human preference (Wang et al., 2023d). Achieving alignment with human preference through SFT necessitates collecting a broad range of training data which is typically gathered through crowd-sourcing (Ouyang et al., 2022; Bai et al., 2022a; Touvron et al., 2023b) or by distilling from other LLMs (Taori et al., 2023; Ding et al., 2023). Recent research indicates that such training data for alignment should be diverse and complex, covering various domains, tasks,

---

[\*]Equally Contributed. Order determined by swapping the one in Yuan et al. (2023a).
[†]Work done during internships at Alibaba DAMO Academy.

semantics, and formats (Xu et al., 2023a; Mukherjee et al., 2023; Wang et al., 2023b). Such diversity and complexity are mainly determined by the query formation. Various methods are proposed and claimed to improve the diversity and complexity of the queries and advance the performance of the SFT-aligned LLMs (Wang et al. 2023c; Xu et al. 2023a; Ding et al. 2023; *inter alia*). However, how to quantify the diversity and complexity of queries is significantly understudied.

To shed light on this topic, we propose using a tagging system to feature and categorize samples in SFT datasets. Given the versatile tasks that the aligned LLMs are expected to handle, an equally versatile tag system is necessary to distinguish open-world human queries. However, building an open, fine-grained tagging system manually is infeasible to scale for large datasets. To this end, we propose INSTAG, an automatic INStruction TAGging method empowered by proprietary high-performing chatbot ChatGPT. Leveraging such a well-aligned chatbot, INSTAG designs a framework to automatically prompt ChatGPT to assign tags to training sample queries. INSTAG achieves the increased quality of the tag assignment by deliberately prompting ChatGPT to explain each tag assigned and including a systematic tag normalization procedure. We apply INSTAG to an extensive collection of open-source SFT datasets and build open-set, fine-grained tags which, as we observed, can reflect the semantics and intentions beneath human queries in SFT datasets. Through the scope of tags, we conduct a detailed and quantified analysis of existing open-source datasets, providing insights into query distributions in terms of diversity and complexity. Such analyses reveal that diverse and complex queries induce high alignment performance through SFT. Following this insight, we propose a data selector based on INSTAG, including a complexity-first diverse sampling method that can extract the most complex queries in a diverse distribution. LLMs fine-tuned with data selected by the INSTAG selector perform well on the popular benchmark MT-Bench (Zheng et al., 2023), supporting our previous query distribution insights.

The contributions of this work are mainly three-fold. Firstly, we propose using open-set fine-grained intention tags as instruction diversity and complexity metrics. To this end, we develop INSTAG, an automated annotator that leverages the instruction-following abilities of proprietary chatbots and employs tag normalization methods. Secondly, we analyze existing open-source SFT datasets and provide insights into query diversity and complexity. Finally, we design a data selector based on INSTAG and apply it to the latest open-source datasets. The resulting best LLMs, TAGLM-13b-v1.0 and TAGLM-13b-v2.0 respectively based on LLaMA (Touvron et al., 2023a) and LLaMA-2 (Touvron et al., 2023b), trained with selected data achieve scores of 6.44 and 6.55 on the benchmark MT-Bench, respectively, surpassing a group of LLMs aligned with considerably more SFT data. Our contributions are verified with rich experiments and multifaceted analysis. Most notably, INSTAG exhibits its robust potential to offer deeper insights into LLMs alignment, extending beyond the data selection introduced in our work.

## 2 RELATED WORKS

**LLMs with Human Alignment.** Through supervised fine-tuning (SFT), rejection sampling, or reinforcement learning (Ouyang et al., 2022; Bai et al., 2022a;b; Yuan et al., 2023b; Rafailov et al., 2023; Song et al., 2023; Touvron et al., 2023b), LLMs can obtain versatile abilities for understanding and following diversified human queries expressed in natural languages, aligning with human intentions. Recent research mainly focused on SFT to align LLMs with human intentions and has contributed essential practices to developing open-resourced well-aligned LLMs, which is adequately summarized by Zhao et al. (2023). Several prominent works collected SFT data through human annotated demonstrations (Ouyang et al., 2022; Touvron et al., 2023b), online user logs of proprietary LLMs (Chiang et al., 2023; Wang et al., 2023a; Köpf et al., 2023), or prompting proprietary high-performing LLMs (OpenAI, 2023) to generate or rewrite samples (Taori et al. 2023; Ding et al. 2023; Xu et al. 2023a; Mukherjee et al. 2023; *inter alia*). Different LLMs fine-tuned on the datasets have aligned with human preference and exhibited good performance in various real-world tasks.

**Data for Human Alignment.** It has been highlighted that the performance of aligned LLMs is affected by the quality of the SFT data. Such data quality pertains to the level of responses (Peng et al., 2023; Chiang et al., 2023), the difficulty of tasks presented (Mukherjee et al., 2023), the complexity of queries (Xu et al., 2023a), the diversity of semantics (Ding et al., 2023; Taori et al., 2023), and the scale of sample amounts (Zhou et al., 2023). Taori et al. (2023) used Self-Instruct (Wang et al., 2023c) to generate diversified queries for SFT and Xu et al. (2023a) proposed Evol-Instruct to complexify simple queries for better human alignment. Mukherjee et al. (2023) used propri-

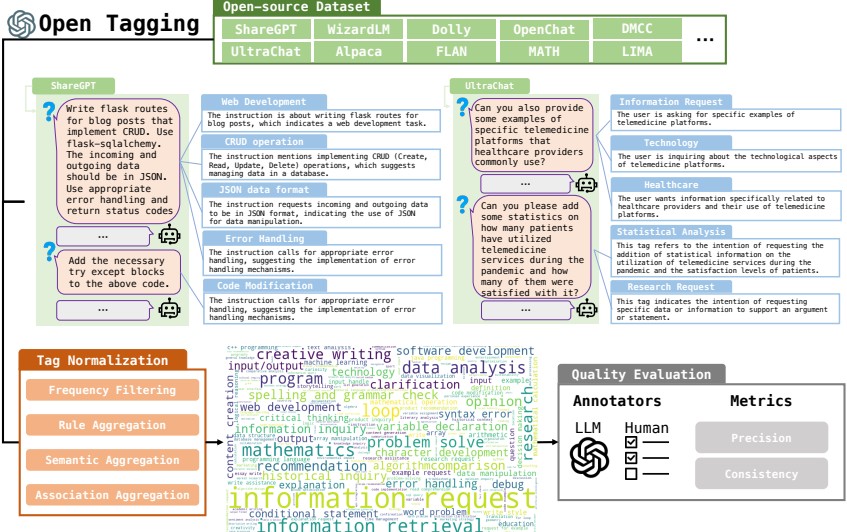

Figure 1: Overview of INSTAG. We use ChatGPT to annotate fine-grained tags for a series of open-source datasets. This figure presents two cases of open tagging annotations from ShareGPT and UltraChat. A tag normalization, including multiple denoising and aggregation methods, is then applied to the original tagging results. Finally, the quality of the tag set, as shown in the word cloud, is evaluated by human and LLM annotators, focusing on the tagging precision and consistency.

etary high-performing LLMs to rewrite the queries and responses of samples from FLAN collection (Longpre et al., 2023) and observed improvement of LLMs in conventional NLP task solving. Ding et al. (2023) proposed UltraChat using manually designed diverse anchor concepts and entities to generate multi-turn data by inducing conversations in ChatGPT. OpenChat (Wang et al., 2023a) and Vicuna (Chiang et al., 2023) are both current open-sourced LLMs with cutting-edge instruction following abilities, and both models are trained on the user logs of GPT-4 from ShareGPT. As evaluated in Wang et al. (2023b), the success of fine-tuning on ShareGPT demonstrates that queries from user logs are of higher diversity and the responses generated from GPT-4 are of better quality, resulting in superior instruction following the abilities. Zhou et al. (2023) found that a small amount of high-quality data is sufficient for LLMs to excel at human alignment.

Although current research proposed more diversified and complex SFT data and made significant progress in developing well-aligned LLMs with human intentions, existing works have yet to discuss how to quantify the diversity and complexity of queries. Taking advantage of the high-performing ChatGPT, we annotate existing data samples with tags. We quantify the diversity and complexity of the training data for the first time and study the data mixture for better alignment.

## 3 INSTAG

This section presents an automatic instruction tagging method INSTAG and its preliminary analyses. We first define fine-grained intention tags and present the open tagging process with LLMs (§3.1). Then, we design a systematic normalization method to denoise the raw tags from previous annotations (§3.2). We also fully evaluate the tagging quality to ensure INSTAG generates precise and consistent intention tags (§3.3). Finally, we use INSTAG to analyze open-source SFT datsets (§3.4).

### 3.1 OPEN-SET FINE-GRAINED TAGGING

Instructions, or queries in prompting modern chatbots, serve as expressions of user intentions, which can often be multifaceted and highly intricate. To illustrate, we showcase an instruction from the ShareGPT dataset in Fig. 1, where the user submits a coding request specifying desired output formats and error-handling methods. To better parse such instructions, employing fine-grained tags to identify fine-grained intentions rather than relying on generalized, coarse-grained classes is essential. However, although fine-grained intention tags offer a more detailed understanding of instruction distribution, they also present challenges in annotation and normalization. Therefore, we propose

| Inconsistency | Examples | Output |
|---|---|---|
| Lexical Noise **Rule Aggregation** | Information Retrieval, information_retrieval, infomation retrieve | information retrieval |
| Uncontrolled Granularity **Semantic Aggregation** | information request, request for information, request for additional information, request for more information, additional information request, specific information request | information request |
| Spurious Correlations **Association Aggregation** | (mathematics, math problem), (loop, for loop) | mathematics, for loop |

Table 1: Inconsistency in intention tagging results from open-set annotations. Inconsistencies can be addressed with three aggregation methods described in §3.2.

an open-set tagging with ChatGPT without providing a predefined ontology of tags and a normalization technique to address these issues. We prefer an open setting since a closed set is not flexible enough to cover versatile intentions in open chatting. Our prompt for tagging is shown in Tab. 5. We provide few-shot examples in the prompt to hint ChatGPT to provide tags in a specific JSON format for accurate parsing. As shown in Fig. 1, we separately annotate each query in a chat session and require ChatGPT to explain tags for the convenience of quality evaluation briefly.

## 3.2 Tag Normalization

The production of intention tags through ChatGPT in an open setting presents a challenge in ensuring consistency, as no predefined ontology is provided, resulting in noise in the raw tagging outcomes. The number of original raw tags for open-sourced datasets is over 12,000, showing ChatGPT can provide diverse and fine-grained annotations. However, we notice the original tagging results contain noticeable noises, including inconsistent word format and granularity. Therefore, we design a systematic method to normalize the open-set tagging. We have identified three significant types of noise, detailed in Tab. 1: **Lexical Noise**, arises from the instability of ChatGPT in adhering to output format instructions and can be mitigated through stemming as a post-processing step; **Uncontrolled Granularity** refers to the potential for ChatGPT to produce overly specific tags; **Spurious Correlations** refer to tags that often appear together due to the bias of ChatGPT or data distributions. Such tag groups should be merged to form an atomic tag. These issues must be addressed to ensure that intentions are accurately identified and utilized in downstream processes. Therefore, we normalize open-set tagging results by various aspects, including frequency, format, semantics, and associations. Specifically, we clean the raw tagging with the following normalization procedure:

- **Frequency Filtering**: We first filter out long-tail tags appearing less than $\alpha$ times in the whole annotated dataset. $\alpha$ is a hyperparameter related to the scale of the dataset.
- **Rule Aggregation**: We transform all tags into lower characters to avoid the influence of capitalization and replace all special characters with spaces. Finally, we apply stemming to each tag with the support of NLTK (Bird et al., 2009).
- **Semantic Aggregation**: We employ text embedding models to obtain the semantics of tags. We use PHRASEBERT (Wang et al., 2021), a BERT-based model designed explicitly for embedding phrases, such as titles of tags. Other embedding methods, such as OpenAI embeddings or DENSEPHRASE (Lee et al., 2020), can also be adopted as alternatives. After obtaining the semantic embeddings of tags, we use DBSCAN algorithm (Hahsler et al., 2019) to cluster tags with a given threshold $t$ of semantic similarity. Similarly, other density clustering methods can be used instead of DBSCAN for the same denoising purpose. Semantic aggregation controls the granularity of tags in terms of semantic similarities.
- **Association Aggregation**: We notice ChatGPT tends to provide highly associated tags that are expected to be considered as an atomic tag as a whole, mainly occurring in mathematics and coding queries. Therefore, we analyze all raw tagging results and employ the FP-Growth algorithm (Han et al., 2000) to mine association rules between tags. We then recursively merge associated tags based on the above association rules and reduce verbosity.

We apply INSTAG on 17 widely-used open-source SFT datasets introduced in Appx. §E. Over 100 thousand original unique tags are generated following the ChatGPT annotation. To filter out long-

| Metric | GPT-4 Annotation | Human Annotation (1%) | Agreement ($\kappa$) | |
| --- | --- | --- | --- | --- |
| | | | Human-Human | Human-GPT |
| **Tag Precision** | 96.1 | 100 | 0.47 | 0.92 |
| **Tag Consistency** | 86.6 | 100 | 0.73 | 0.75 |

Table 2: Evaluation for the tagging quality of INSTAG. We design two metrics, tagging precision and consistency, for evaluating INSTAG. Moreover, we also employ three human annotators to annotate 1% cases and report their majority voting. We report agreement between human annotators in Fleiss-kappa scores and agreement between majority voting and GPT-4 in Cohen's kappa scores.

tail cases, we implement Frequency Filtering with $\alpha = 20$, resulting in the retention of 8,541 tags. We apply the rule aggregation to address lexical noise, which reduces tags to 7,157. Then, semantic aggregation with a minimum semantic similarity 0.05 reduces the count to 6,587 tags. Finally, we employed the association aggregation with a minimum support of 40 times and a minimum confidence of 99%, producing 1,772 association rules to transform tag groups into atomic tags. These measures were essential in streamlining the tagging process and ensuring the quality of downstream processes. An overview of frequent tags is in Appx. §B. We also train a local specialized tagging LLM, INSTAGGER, to distill such annotation abilities into smaller LLMs, shown in Appx. §C.

## 3.3 QUALITY EVALUATION

We employ both GPT-4 and human annotators to provide judgments in tagging quality. The quality of the normalized tagging dataset is evaluated in precision and consistency:

- **Precision** Precision is whether tags assigned to a specific query correctly relate to query intentions. Tag precision is essential since fine-grained tags should be precisely expressed as part of query intentions. For example, given a case $(q, \mathcal{T})$ where $q$ is the query and $\mathcal{T}$ is tags assigned to it, we employ annotators to identify any incorrect tags in $\mathcal{T}$. We consider it a negative case if any tag in $\mathcal{T}$ is annotated as incorrect. Otherwise, it is a precise tagging case.

- **Consistency** To form a consistent tag ontology, we naturally require that the semantics of a specific tag will not shift across queries. An annotation case in consistency $(t, \mathcal{I})$ contains a tag $t$ and a set of randomly selected instructions $\mathcal{I}$ annotated with such tag. Annotators are required to identify any semantic changes in tags across all instructions.

Specifically, we randomly sample 4,000 cases for GPT-4 annotation, 2,000 each for precision and consistency. Then, we hire three annotators to manually label 40 cases (1%) selected from the above set. Manual annotations provide judgments and reveal confidence of GPT-4 annotation. The evaluation results are shown in Tab. 2. GPT-4 provides 96.1% and 86.6% accuracy in tag precision and consistency, respectively. Meanwhile, we also report the majority voting of human annotators, which suggests a hundred percent correctness among both tasks. We notice the Fleiss-kappa between human annotators reaches the basic agreement. In contrast, Cohen's kappa between majority voting and GPT-4 reaches more than 0.7, suggesting a solid agreement between human and GPT-4 annotators. To eliminate the possibility that such results contain robust false positive annotations, we specifically design counterfactual annotation experiments shown in Tab. 9 and proof that both human and GPT-4 are capable of precisely recalling incorrect cases. Therefore, tags provided by INSTAG are of good quality regarding precision and consistency for downstream analyses.

## 3.4 PRELIMINARY ANALYSIS

We present the analysis of open-source datasets through normalized tags in Fig. 2. To start with, we introduce the diversity and complexity attributes of SFT datasets induced by tagging results:

- **Diversity** is used to access the range of intentions and semantics covered by queries in a dataset. According to the tagging results, a dataset is considered more diverse if it covers more individual tags. The attribute is quantified as the unique tag coverage rate for the overall tag set.

- **Complexity** aims to measure the number of intentions complicating queries. We assume a more complex query would be assigned more tags. The attribute is quantified as the average tag number assigned to queries in a dataset.

We first depict the overall assessments regarding the axis of diversity and complexity as shown in Fig. 2a. Each dataset is represented as a dot whose size indicates the sample size, and color indicates

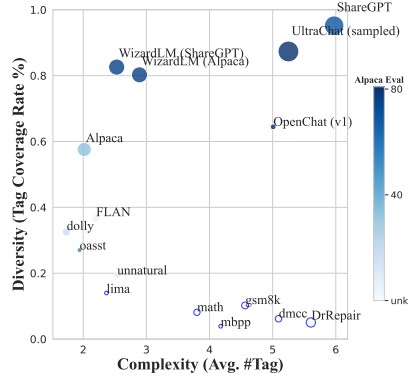 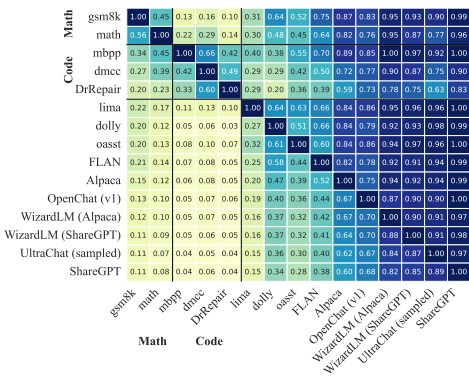

(a) Diversity and Complexity based on Tags (b) Dataset Correlation (Column recalls Row)

Figure 2: Dataset analysis based on tags. Fig. 2a shows diversities and complexities based on tags, where data scales and AlpacaEval scores are marked in dot sizes and colors respectively. Datasets without AlpacaEval scores are marked in circles. Fig. 2b shows correlations among datasets based on the recalls of tags. Numbers are recalls using tags in the column against tags in the row.

the performance of LLMs fine-tuned thereon and tested on AlpacaEval (Li et al., 2023). As shown, (1) **Tag-based metrics well presents diversity and complexity.** WizardLM (Alpaca) is created by complicating the queries from Alpaca datasets using Evol-Instrcut (Xu et al., 2023a). We can see that WizardLM (Alpaca) has a larger coverage rate and average tag number than the Alpaca dataset. This observation demonstrates the complexity and diversity of an SFT dataset can be well quantified by tags. (2) **The larger size, the more diverse and more complex.** On both axes, the larger datasets naturally contain human queries of higher diversity and complexity, except for mathematical reasoning and code generation. (3) **Math and Code show different trends.** The datasets for mathematical reasoning (MATH (Hendrycks et al., 2021), GSM8K (Cobbe et al., 2021)) and code generation (DMCC (Li et al., 2022), MBPP (Austin et al., 2021), DrRepair (Yasunaga & Liang, 2020)) focus on specific downstream abilities and result in low diversity, while such datasets have relatively high complexity. (4) **Diverse and complex data induces higher performance.** ShareGPT, UltraChat (Ding et al., 2023), and OpenChat-v1 (Wang et al., 2023a) datasets lay at the upper-right corner of Fig. 2a, having both high diversity and complexity. Vicuna (Chiang et al., 2023), UltraChat, and Openchat, respectively fine-tuned on the datasets, achieve cutting-edge performance among open-sourced models, as evaluated by public leaderboards (e.g., AlpacaEval). This scenario verifies that LLMs can benefit from fine-tuning more diverse and complex data for alignment.

We display the correlations between datasets regarding tag recalls to understand the correlations between open-source SFT datasets. As depicted in Fig. 2b, we use the tag sets of the datasets on each column to calculate the recall to the tag sets of the datasets on each row. We can conclude from the figure that (1) **Tags can identify different tasks.** Datasets for mathematical reasoning and code generation tasks exhibit higher tag recalls within the tasks. This demonstrates that the tags can identify the uniqueness of mathematical reasoning and code generation datasets compared to more general-purpose datasets. (2) **Few cover all.** WizardLM (Alpaca), WizardLM (ShareGPT), UltraChat, and ShareGPT have higher tag recalls for other datasets. These four datasets contain more diversified queries and cover other datasets, consistent with the results in Fig. 2a.

Overall, INSTAG provides a tool for analyzing SFT datasets through the perspective of tagging. Existing SFT datasets differ in diversity and complexity as evaluated by the tagging results. However, we also notice two outliers in these figures. The dataset for Alpaca seems to have a large data size while resulting in inferior performance and low complexity. The dataset for OpenChat-v1 is filtered from ShareGPT, resulting in high query diversity and complexity while having only 8K multi-turn conversations, which suggests a considerably small data scale with high query complexity and diversity can potentially result in better performance. We give more analysis on data size in §4.3.

## 4 INSTAG FOR DATA SELECTION

As analyses in §3.4, we notice fine-tuning LLMs on more diverse and complex datasets may benefit alignment performance. Therefore, we present a data selection method supported by INSTAG in this

| Model | Data Size | MT-Bench | AlpacaEval |
|---|---|---|---|
| **Proprietary Models** | | | |
| GPT-4 | − | 8.99 | 95.3 |
| GPT-3.5-turbo | − | 7.94 | 86.1 |
| Claude-v1 | − | 7.90 | 88.4 |
| **LLaMA-2 Based Open-source Models** | | | |
| Llama-2-13b-chat (Touvron et al., 2023b) | − | 6.65 | 81.1 |
| **TAGLM-13b-v2.0** | 6K | $6.55_{\pm 0.02}$ | $80.9_{\pm 1.4}$ |
| **LLaMA Based Open-source Models** | | | |
| Alpaca-13b (Taori et al., 2023) | 52K | 4.53 | 21.9 |
| OpenChat-13b-v1 (Wang et al., 2023a) | 8K | 5.22 | 80.9 |
| Baize-v2-13b (Xu et al., 2023b) | 56K | 5.75 | 67.0 |
| Vicuna-13b-v1.1 (Chiang et al., 2023) | 70K | 6.31 | 70.4 |
| WizardLM-13b (Xu et al., 2023a) | 70K | 6.35 | 75.3 |
| Vicuna-13b-v1.3 (Chiang et al., 2023) | 125K | 6.39 | 82.1 |
| **TAGLM-13b-v1.0** | 6K | $6.44_{\pm 0.04}$ | $75.8_{\pm 1.5}$ |

Table 3: Main results of TAGLM. Standard deviations are derived under three GPT-4 judgments and obtain results for other baselines from the official MT-Bench and AlpacaEval leaderboard. Dashes in the data column denote unknown data sizes. Detailed results are presented in Tab. 8.

section and align LLMs with selected data to show the effectiveness of INSTAG. We introduced experimental setup (§4.1), results (§4.2), and analyses related to query diversity and complexity (§4.3).

## 4.1 EXPERIMENTAL SETUP

**Data Pool.** Based on the normalized tagging results and the preliminary analyses of open-source datasets as presented in Figure 2, we conduct fine-grained experiments to discuss the impact of data complexity and diversity through controlling-for-a-variable methods. Under the correlation analyses in Figure 2b, each dataset of WizardLM(Alpaca), WizardLM(ShareGPT), UltraChat, and ShareGPT maintains large tag recalls regarding other datasets. The four datasets also have the largest average tag numbers shown in Figure 2a. These results indicate that the four datasets have high coverage for other datasets regarding tags. Therefore, we pool the four datasets and create different subsets for data complexity and diversity analysis. The pooled dataset contains 306,044 samples with a tag set size 6,398 and an average tag number of 4.48. Detailed datasets are in Appx. §E.

**Configuration.** We use the dataset of 6K samples to align the 13B version of LLaMA (Touvron et al., 2023a) and LLaMA-2 (Touvron et al., 2023b) with human preference via SFT, and dub both aligned LLMs TAGLM-13b-v1.0 and TAGLM-13b-v2.0 respectively. All the models fine-tuned in the following analyses are based on 13B version LLMs of either LLaMA (Touvron et al., 2023a) or LLaMA-2 (Touvron et al., 2023b). If not specified otherwise, we fine-tune the model for five epochs with the batch size set to 128 and the learning rate set to $2 \times 10^{-5}$. The Vicuna-style template is applied to concatenate queries and responses during fine-tuning. We evaluate each fine-tuned model on MT-Bench (Zheng et al., 2023) strictly following its recipe using GPT4 as a judge to demonstrate the alignment performance, set comparison to other LLMs, and conduct analyses.

**Data Sampling.** LLMs can benefit from datasets with higher diversity and complexity according to the analyses in §3.4. We sample a data subset of 6K samples from the pooled dataset with the highest sample complexity of an average tag number 16.56 and tag coverage of 100%. We propose a **Complexity-first Diverse Sampling** procedure (Cf-D, Alg. 1) to obtain the datasets.

**Baselines.** We compare our models to two sets of baselines. We first use proprietary GPT-4, GPT-3.5, and Claude-V1 as strong baselines, and then include strong cutting-edge open-sourced aligned LLMs, Vicuna (Chiang et al., 2023), WizardLM (Xu et al., 2023a), Baize (Xu et al., 2023b), Open-Chat (Wang et al., 2023a), and Alpaca (Taori et al., 2023). Details are left to Appx. §F.

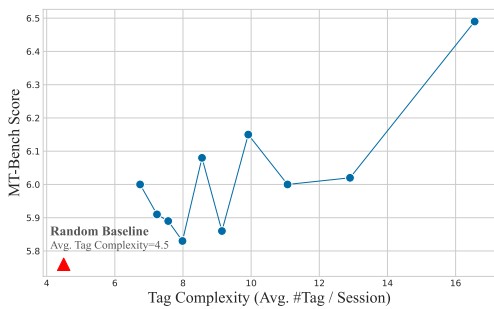
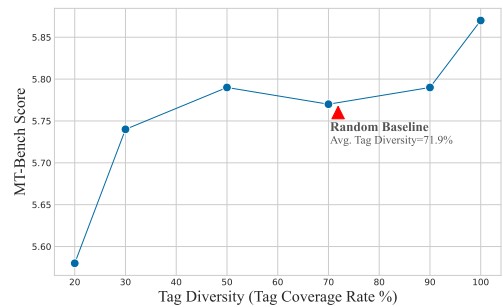

(a) Performance under different Tag Complexities          (b) Performance under different Tag Diversities

Figure 3: Analysis results of model performance in terms of different tag complexities and diversities. Fig. 3a shows MT-Bench scores over different tag complexities defined as an average number of tags per session. Fig. 3b shows scores over different tag diversities defined as coverage rates over all tags. We include a random baseline in both figures as shown in red triangles.

## 4.2  RESULTS

As shown in Tab. 3, TAGLM-13b-v1.0 outperforms all the open-sourced aligned LLMs, achieving a 6.44 average score on MT-Bench, although it is only fine-tuned based on LLaMA on 6K samples, far less than those of other LLMs. We report the average of three GPT-4 judgments and provide the standard deviation of scores as we notice randomness in GPT-4 judgments. This result illustrates that diversity and complexity matter in human alignment by SFT. Our INSTAG provides a decent tool for accessing and quantifying both attributes. TAGLM-13b-v2.0 fine-tuned based on LLaMA-2 achieves even higher results while lagging behind LLaMA-2-chat by only 0.1, which is aligned with human preference via RLHF. Compared to proprietary high-performing LLMs, especially GPT-4, the performance falls behind on MT-Bench. We also present more detailed results on MT-Bench in terms of eight tasks in Appx. §G. We also report evaluation results on AlpacaEval and we witness a similar conclusion as MT-Bench. TAGLM-13b-v1.0 outperforms most of the baselines that are aligned with much more data and achieves a 75.8 win rate compare with *text-davinci-003*, but falls behind OpenChat-13b-v1 and Vicuna-13b-v1.3. TAGLM-13b-v2.0 also achieves comparable performance on AlpacaEval with Llama-2-13b-chat.

## 4.3  DECOUPLED ANALYSIS

We primarily discuss how SFT data size relates to the alignment performance and give an ablation study on the sampling procedure according to tags. The results are shown in Tab. 4. Using the sampling procedure in Alg. 1, the alignment performance achieves the best score with 6K data, and the performance degrades when the data size increases to 10K and 16K. As compared to SFT with all and half of the whole pooled data, the performance remains superior. These results empirically verified there exists a small-scale subset with high diversity and complexity that can lead to excellent alignment performance. The finding is consistent with LIMA (Zhou et al., 2023). Comparing Alg. 1 to random sampling with the same 6K samples, the proposed sampling procedure results in significantly better performance than random sampling, largely surpassing by 0.68 on MT-Bench.

| Selection | Data Size | MT-Bench |
|---|---|---|
| Cf-D | 3K | 5.92 |
| | 5K | 6.33 |
| | 6K | 6.44 |
| | 10K | 6.34 |
| | 16K | 6.31 |
| Random | 6K | 5.76 |
| | 10K | 6.27 |
| | 153K(half) | 6.23 |
| - | 306K(all) | 6.21 |

Table 4: Results for different SFT data sizes and sampling procedures. Cf-D represents the complexity-first diverse sampling in Alg. 1.

We then provide decoupled analyses of complexity and diversity to demonstrate how they influence alignment performance given the same SFT data size.

**Complexity.** To decouple and focus on the data complexity, we sample different data subsets of diverse averaged tag numbers. Different sampled subsets share the same sample size of 6k and have

the same tag coverage of 100%, implying the largest data diversity. In the sampling procedure, all the data samples are first sorted by the tag numbers in descending order. Then for each data subset, we start from the sample in the whole dataset with the largest tag number. The sample that can expand the tag set size of the current sampled data will be extracted and removed from the whole dataset. If the tag set of the current sampled subset covers the whole tag set and the sample number is still less than 6k, we repeat the sampling procedure until the sample numbers reach 6k. This sampling procedure is similar to Alg. 1. We leave the detail in Appx. §H.

We sample 10 different data subsets, and the average tag numbers of the subsets range from 6.7 to 16.6. As shown in Figure 3a, the overall performance trend on MT-Bench is increasing along with the growth of average tag numbers. This trend may not be significant on the fine-grained level of average tag numbers where the number difference between subsets is small. Compared to the randomly sampled datasets, the average tag number of around 4.5, all the 10 data subsets can lead to superior fine-tuned model performance than the randomly sampled subset baseline. To summarize, on a coarse-grained level of data complexity, the downstream performance is positively correlated to the average tag number, while on a fine-grained level, such a phenomenon becomes less evident. This may be partly because ChatGPT does not recall all the possible tags for each query, or some tags are filtered out during the tag normalization procedure, resulting in a less accurate tag number.

**Diversity.** For diversity, we sample different data subsets spanning various tag coverage rates regarding the whole tag set. Different subsets share the same sample scale of 6k and the same average tag number, implying the same data complexity. The average tag number is set to 5.0. For data subset sampling, we first draw samples that can expand the tag set size of the current sampled data until the target tag coverage rate. Then, we traverse the remaining samples and extract samples that do not expand the tag coverage and can keep the current average tag number of the subset around 5.0. We leave the detailed sampling algorithm for diversity analysis in Appx. §H.

We can observe in Figure 3b that as the tag coverage increases, the fine-tuned model can achieve higher MT-Bench scores. Randomly sampled data subsets of tag coverage 71.9% result in similar model performance with the sampled subset of tag coverage 70%. This demonstrates that the fine-tuned models may benefit from the more diverse datasets through the scope of tags. The trend is not strictly linear, and there seems to be a plateau ranging from 50% to 90% coverage. This could be due to the tags assigned may not share the sample importance for diversity.

## 5    INSTAGGER: LOCAL TAGGER BY DISTILLATION

INSTAG depends on advanced chatbots that are expensive on large-scale applications. As fine-grained tags benefit SFT data selection and other applications, we naturally propose INSTAGGER, which is equipped with the tagging ability of these high-performing chatbots by distillation of de-noised tagging results with significantly fewer budgets and higher efficiency. Distilling is an effective method to inject a smaller model with specialized abilities (Fu et al., 2023). We use our INSTAG results on open-sourced SFT datasets to fine-tune a 7B version LLaMA-2 model. The detailed implementation is described in Appx. §C. We validate the model on our validation set. The tag-level F1 score based on exact match (EM) and semantic-based fuzzy match are 31.8% and 73.4%. As this is an unconstrained open-generated tagging, EM is a rigorous metric for annotating over six thousand tags. Therefore, we also calculate the fuzzy match by PhraseBERT, which considers a predicted tag is correct if it has over 0.8 cosine similarity in semantics with any gold tag.

## 6    CONCLUSION

In this paper, we introduced INSTAG, an open-set tagging method leveraging the instruction-following ability of ChatGPT for SFT data analysis. We apply INSTAG on open-source SFT datasets, showing diverse and complex data leads to better alignment performance. We designed a complexity-first diverse sampling method to select 6K samples, and TAGLM fine-tuned on this selected dataset outperforms other open-source models aligned with considerably more data. Moreover, further decoupled analyses revealed that model performance increases with fine-tuning on more diverse and complex SFT data, respectively. In summary, our proposed INSTAG provides a novel aspect for a deeper understanding of query distribution in the alignment of LLMs. It has robust potential to be extended to more applications beyond the data selection shown in this work, such as creating comprehensive evaluations and tag-based self-instruct.

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

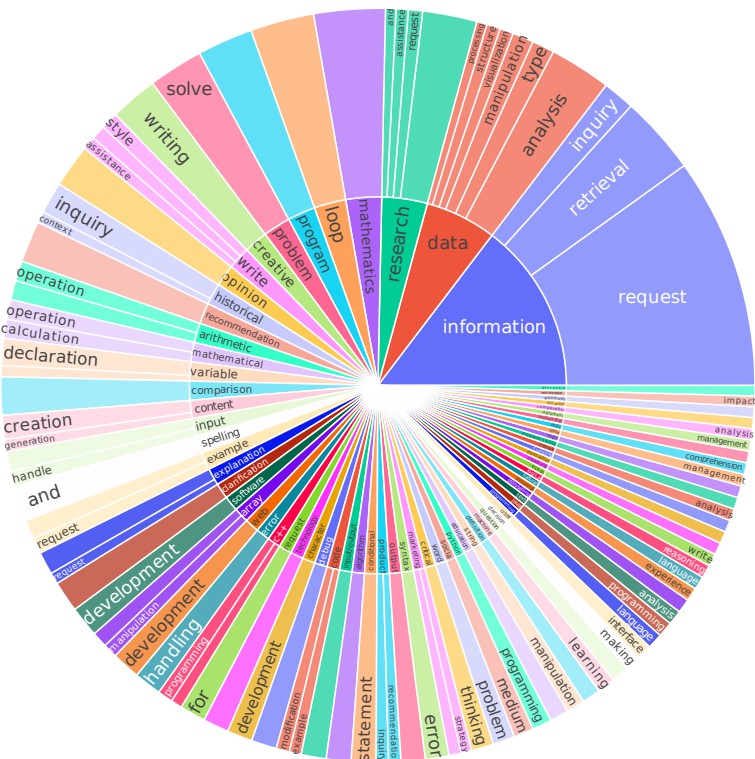

Figure 4: The sunburst plot of all tags. We plot with the first two words of each tag and the size is proportional to the frequency of the tag.

## APPENDIX

## A    LIMITATIONS

Our conclusions mainly rely on MT-Bench and AlpacaEval for model evaluations, which may miss some influence caused by SFT data. Besides, we notice MT-Bench shows instabilities in terms of the randomness of GPT-4 judgments, so we provide random ablations as comprehensively as possible to show the statistical significance of our results, including reporting standard variance of MT-Bench scores. Furthermore, our analysis of SFT datasets is mainly focused on English, so our claims may not be directly extended to multi-lingual scenarios.

## B    TAG REVIEW

We present a sunburst plot of all tags in Fig. 4 showing the most frequent tags is about information-related, data manipulations, and coding queries. We plot with the first two words of each tag and the size is proportional to the frequency of the tag. We only plot with tags that have frequencies larger than 2000 in our data pool.

## C    LOCAL TAGGER

We use the following template to concatenate queries to tag and tagging results:

> You are a helpful assistant. Please identify tags of user intentions in the following user query and explain each tag. Please respond in the JSON format {"tag": str, "explanation": str}. Query: <query-to-tag> Assistant: <tagging-results>

| |
|---|
| You are a tagging system that provides useful tags for instruction intentions to distinguish instructions for a helpful AI assistant. Below is an instruction:
[begin]
{instruction}
[end]
Please provide coarse-grained tags, such as "Spelling and Grammar Check" and "Cosplay", to identify main intentions of above instruction. Your answer should be a list including titles of tags and a brief explanation of each tag. Your response have to strictly follow this JSON format: [{"tag": str, "explanation": str}]. Please response in English. |

Table 5: ChatGPT prompt template for annotating intention tags of given queries.

| |
|---|
| You are an experienced judge for intention tags of instructions. You will be provided a query and a list of tags describing intentions of the query as followed:
[query]: {query}
{tags}
Please provide feedback about whether all tags precisely describe an intention of the instruction. Please identify all incorrect tags and provide their indices in the JSON format as your response. The JSON format for your response is a list of JSON dictionary and the JSON dictionary has only one key to identify the index of each incorrect tag: [{"idx": int}]. For example, if [tag 0] and [tag 2] are incorrect, you should response as [{"idx": 0}, {"idx", 2}]. If all tags are correct, please response an empty list as []. |

Table 6: GPT-4 prompt template for evaluating tagging precision.

We also include the explanation in the tagging results to make the fine-tuned model obtain better tagging performance. The overall sample size for fine-tuning is 773,511, where we randomly sample 1,000 samples for validation. The model is fine-tuned with 512 batch size for one epoch since we empirically find that training for more than one epoch will lead to over-fitting.

## D  PROMPT TEMPLATES FOR CHATGPT

We preset our prompt for ChatGPT for annotation (Tab. 5), precision evaluation (Tab. 6), and consistency evaluation (Tab. 7).

## E  DATASETS

We apply INSTAG to 17 open-source SFT datasets for intention tagging:

- **ShareGPT**[1] refers to the multi-turn chatting histories used by VICUNA (Chiang et al., 2023). ShareGPT includes human-written queries and responses from ChatGPT and other chatbots.
- **OpenChat** (Wang et al., 2023a) is a subset of ShareGPT containing only chat histories with GPT-4 responses.[2]

---

[1]Exact dataset of ShareGPT (https://sharegpt.com/) has not been released. We instead use a reproduced version from https://huggingface.co/datasets/anon8231489123/ShareGPT_Vicuna_unfiltered/tree/main/HTML_cleaned_raw_dataset, and follow Vicuna preprocess.

[2]We use the dataset with 8,000 GPT-4 responses denoting as OpenChat v1.0 in https://huggingface.co/datasets/openchat/openchat_sharegpt4_dataset

---

You are an experienced judge for consistency of intention tags for instructions. You will be provided a tag and a list of instructions labeled with this tag as followed:
[tag]: {tag}
{instructions}
Please provide feedback about whether the meaning of this tag is consistent among all instructions. Please identify all inconsistent instructions and provide their indices in the JSON format as your response. The JSON format for your response is a list of JSON dictionary: [{"idx": int}]. For example, if the meaning of tags in [instruction 0] and [instruction 2] are inconsistent, you should response as [{"idx": 0}, {"idx": 2}]. If the meaning of tag is consistent in all instructions, please response an empty list as [].

---

Table 7: GPT-4 prompt template for evaluating tagging consistency.

- **UltraChat** (Ding et al., 2023) is a systematically designed, diverse, informative, large-scale dataset of multi-turn instructional conversations without involving human queries.[3]
- **Alpaca** (Taori et al., 2023) is a dataset generated by the modified SELF-INSTRUCT method (Wang et al., 2022), containing 52,000 instruction-following demonstrations generated from OpenAI's *text-davinci-003* model.[4]
- **WizardLM** (Xu et al., 2023a) is an instruction dataset built with the EVOL-INSTRUCT method. EVOL-INSTRUCT utilizes ChatGPT to augment the complexity of the same queries in Alpaca and ShareGPT. We denote these two subsets as WizardLM(Alpaca) and WizardLM(ShareGPT) for clarification.[5]
- **FLAN** (Wei et al.) is a series of data from NLP tasks formatted in instruction tuning. The queries in FLAN are generated by templates for each NLP task.
- **Dolly** (Conover et al., 2023) contains 15,000 high-quality human-generated prompt and response pairs for instruction tuning of LLMs.
- **OAssist** (Köpf et al., 2023) is a crowdsourced human-annotated dataset about multi-lingual conversations.
- **Unnatural** (Honovich et al., 2022) contains queries generated by prompting DAVINCI-002.
- **Lima** (Zhou et al., 2023) contains only 1,000 carefully human-curated prompts and responses.
- **Math Collections**: We involve a set of math datasets including GSM8K (Cobbe et al., 2021) and MATH (Hendrycks et al., 2021) to prompt INSTAG generating fine-grained mathematical tags.
- **Code Collections**: We also involve a set of code datasets including DMCC (Li et al., 2022), MBPP (Austin et al., 2021), and DrRepair (Yasunaga & Liang, 2020) for the same purpose as introducing mathematical datasets.

## F  BASELINE LLMs

We give introductions to the LLM baselines for human alignment.

- **Alpaca** (Taori et al., 2023) is the first open-sourced LLM aligned with human preference. Alpaca is fine-tuned on SFT data of 52K samples generated from text-davince-003 using Self-Instruct (Wang et al., 2023c).
- **WizardLM** (Xu et al., 2023a) is fine-tuned on the SFT data enhanced with a novel technique named Evol-Instruct. It complexifies the Alpaca SFT data using ChatGPT and achieves better alignment performance.
- **Vicuna** (Chiang et al., 2023) is an aligned LLM fine-tuned on collected user chatting logs of proprietary high-performing chatbots on ShareGPT.

---

[3]https://huggingface.co/datasets/stingning/ultrachat

[4]We collect the Alpaca dataset along with Dolly, OAssist, and Unnatural from the sharing of Wang et al. (2023b)https://github.com/allenai/open-instruct.

[5]We use the V2 version of WizardLM in https://huggingface.co/datasets/WizardLM/WizardLM_evol_instruct_V2_196k.

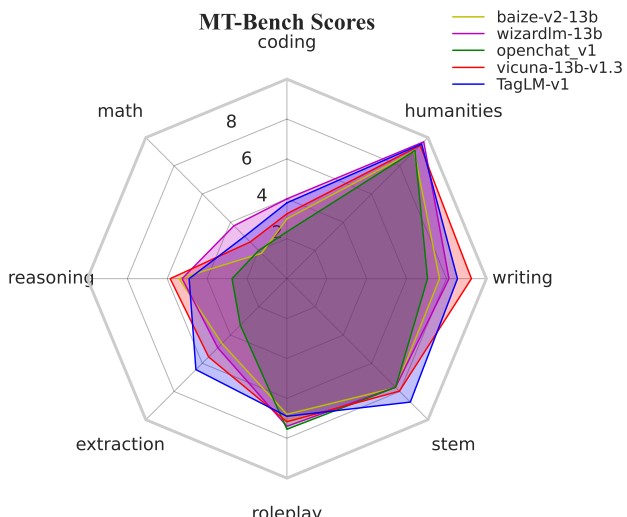

Figure 5: Radar plot showing detailed scores of TAGLM-13b-v1.0 and major baselines on eight subtasks of MT-Bench. Detailed numbers can be viewed in Tab. 8.

- **OpenChat** (Wang et al., 2023a) is fine-tuned on a subset of ShareGPT with only the chatting logs with GPT-4.
- **Baize** (Xu et al., 2023b) uses 100K dialogues generated by self-chatting of ChatGPT. It also includes Alpaca's data for SFT.
- **LLaMA-2 Chat** (Touvron et al., 2023b) differs from the above-mentioned LLMs in (1) being based on per-trained LLaMA-2 instead of LLaMA (Touvron et al., 2023a); (2) being aligned with human preference by both SFT and RLHF.

## G  DETAILED MT-BENCH SCORES IN CATEGORIES

As shown in Fig. 5 (we present our detailed number results in Tab. 8), TAGLM-13b-v1 outperforms all other baselines on *stem* and *extraction*, and achieves comparable performances on *humanities* with Vicuna, suggesting these tasks may rely on few data for alignment. TAGLM-13b-v1 ranks the second on *math*, *coding*, and *writing*, but falls short on *roleplay* and *reasoning*. These detailed results show that some tasks may require diverse but only a few alignment data, while tasks about reasoning and writing may continually benefit from large-scale data.

## H  SAMPLING ALGORITHM FOR DECOUPLED ANALYSIS

We calculate complexity and diversity with tag-based metrics described in §3.4. We first sort all samples by the query complexity (the query tag number) and then pick distinct queries according to tags to achieve high sample diversity (tag coverage). The selection criterion at each time is to select a query with large tag numbers and can increase the tag set of the selected subset data. The algorithm is detailed in Alg. 1.

We present our sampling algorithm for decoupled analysis of complexity and diversity in Alg. 2 and Alg. 3, respectively.

| Model | Data | MT-Bench Scores | | | | | | | | Average | |
|---|---|---|---|---|---|---|---|---|---|---|---|
| | | code | extraction | humanities | math | reason | roleplay | stem | writing | all | w/o C&M |
| gpt-4 | − | 8.55 | 9.38 | 9.95 | 6.8 | 9.0 | 8.9 | 9.7 | 9.65 | 8.99 | 9.43 |
| gpt-3.5-turbo | − | 6.9 | 8.85 | 9.55 | 6.3 | 5.65 | 8.4 | 8.7 | 9.2 | 7.94 | 8.39 |
| claude-v1 | − | 6.25 | 8.8 | 9.7 | 4.8 | 5.95 | 8.5 | 9.7 | 9.5 | 7.9 | 8.69 |
| Llama-2-13b-chat | - | 3.0 | 6.92 | 9.75 | 3.45 | 5.1 | 7.5 | 8.62 | 8.85 | 6.65 | 7.79 |
| TAGLM-13b-v2.0 (1) | 6K | 3.75 | 6.5 | 9.55 | 2.1 | 5.3 | 7.95 | 8.5 | 8.75 | 6.55 | 7.76 |
| TAGLM-13b-v2.0 (2) | 6K | 3.7 | 6.2 | 9.52 | 2.15 | 5.35 | 8.1 | 8.4 | 8.95 | 6.55 | 7.75 |
| TAGLM-13b-v2.0 (3) | 6K | 3.4 | 7.35 | 9.6 | 2.15 | 5.9 | 7.45 | 8.28 | 8.0 | 6.52 | 7.76 |
| TAGLM-v1.0-13b (1) | 6K | 3.8 | 6.45 | 9.55 | 3.0 | 4.9 | 6.9 | 8.75 | 8.55 | 6.49 | 7.52 |
| TAGLM-v1.0-13b (2) | 6K | 3.45 | 6.35 | 9.65 | 2.95 | 4.95 | 7.15 | 8.65 | 8.5 | 6.46 | 7.54 |
| TAGLM-v1.0-13b (3) | 6K | 3.4 | 6.45 | 9.45 | 2.85 | 5.05 | 7.05 | 8.43 | 8.4 | 6.38 | 7.47 |
| vicuna-13b-v1.3 | 125K | 3.25 | 5.55 | 9.45 | 2.6 | 5.85 | 7.18 | 7.98 | 9.25 | 6.39 | 7.54 |
| vicuna-13b-v1.1 | 70K | 2.95 | 6.4 | 9.45 | 2.9 | 4.65 | 7.5 | 8.55 | 8.05 | 6.31 | 7.43 |
| wizardlm-13b | 70K | 4.0 | 4.9 | 9.7 | 3.75 | 5.25 | 7.4 | 7.7 | 8.12 | 6.35 | 7.18 |
| baize-v2-13b | 56K | 3.0 | 4.6 | 9.02 | 1.8 | 5.4 | 6.8 | 7.72 | 7.65 | 5.75 | 6.87 |
| nous-hermes-13b | 300K | 2.45 | 5.05 | 9.0 | 2.65 | 3.8 | 6.38 | 7.02 | 7.75 | 5.51 | 6.5 |
| gpt4all-13b-snoozy | 900K | 3.0 | 4.8 | 8.85 | 1.2 | 4.2 | 7.0 | 6.9 | 7.35 | 5.41 | 6.52 |
| koala-13b | 472K | 2.9 | 4.15 | 8.45 | 1.9 | 4.0 | 6.85 | 7.2 | 7.35 | 5.35 | 6.33 |
| openchat-13b-v1 | 8K | 2.35 | 3.3 | 9.07 | 2.0 | 2.75 | 7.55 | 7.7 | 7.05 | 5.22 | 6.24 |
| alpaca-13b | 52K | 2.35 | 4.15 | 7.85 | 1.05 | 3.5 | 5.45 | 5.2 | 6.7 | 4.53 | 5.48 |

Table 8: Main results of INSTAG. We present MT-Bench scores of both proprietary and open-source baselines in similar scales. We also provide average scores overall categories and categories without code and math (w/o C&M). Dashes in the data column denote unknown data scales. Parentheses mark the three different rounds of GPT-4 judgments.

---

**Algorithm 1:** Complexity-first Diverse Sampling

**Data:** The Whole Pooled Dataset $\mathcal{D}$, Sub-Dataset Size $N$
**Result:** The Sampled Sub-Dataset $\mathcal{D}_s$

1 Initialize Empty $\mathcal{D}_s$;
2 Sorting Queries in $\mathcal{D}$ by tag number in descending;
3 **while** $|\mathcal{D}_s| < N$ **do**
4      Tag Set $\mathcal{T}_s^B \leftarrow \emptyset$;
5      **foreach** *Query* $q \in \mathcal{D}$ **do**
6          **if** *Query Tags* $\mathcal{T}_q : |\mathcal{T}_s^B \cup \mathcal{T}_q| > |\mathcal{T}_s^B|$ **then**
7              $\mathcal{D}_s \leftarrow \mathcal{D}_s \cup \{q\}$;
8              $\mathcal{T}_s^B \leftarrow \mathcal{T}_s^B \cup \mathcal{T}_q$;
9              $\mathcal{D} \leftarrow \mathcal{D} \setminus \{q\}$;
10          **if** $|\mathcal{D}_s|$ *equals to* $N$ **then**
11              Break;

12 **return** $\mathcal{D}_s$

---

# I  COUNTERFACTUAL EVALUATION

To test how well annotators can evaluate tag quality, we created counterfactual cases for two tasks. In the tag precision task, we substituted some tags with similar ones in terms of semantics. In the tag consistency task, we used inconsistent instructions to replace the original instructions. Both humans and GPT-4 are able to recognize most of the counterfactual cases. And humans are better at tag precision, while GPT-4 is better at tag consistency. This analysis shows that annotators have low false positive rates and proof confidence of their judgments in the original tagging results.

---

**Algorithm 2:** Data Sampling for Complexity Analysis

---

**Data:** The Whole Pooled Dataset $\mathcal{D}$

1 , Subset Size $N$ **Result:** The Sampled Sub-Dataset of Different Complexity
$$D = \{\mathcal{D}_c^i | i = 1, \ldots, n\}$$

2 Sorting Queries in $\mathcal{D}$ by tag number in descending;

3 Initialize D = list();

4 **foreach** $i$ *in* $\{1, \ldots, n\}$ **do**

5      Initialize Empty $\mathcal{D}_c^i$;

6      **while** $|\mathcal{D}_c^i| < N$ **do**

7          Tag Set $\mathcal{T}_c^B \leftarrow \emptyset$;

8          **foreach** *Query* $q \in \mathcal{D}$ **do**

9              **if** *Query Tags* $\mathcal{T}_q : |\mathcal{T}_c^B \cup \mathcal{T}_q| > |\mathcal{T}_c^B|$ **then**

10                  $\mathcal{D}_c^i \leftarrow \mathcal{D}_c^i \cup \{q\}$;

11                  $\mathcal{T}_c^B \leftarrow \mathcal{T}_c^B \cup \mathcal{T}_q$;

12                  $\mathcal{D} \leftarrow \mathcal{D} \setminus \{q\}$;

13                  **if** $|\mathcal{D}_c^i| = N$ **then**

14                      D $\leftarrow$ D appends $\mathcal{D}_c^i$;

15                      Break;

16 **return** D

---

---

**Algorithm 3:** Data Sampling for Diversity Analysis

---

**Data:** The Whole Pooled Dataset $\mathcal{D}$, Preset Coverage Rate $\mathcal{R} = \{r^i | i = 1, \ldots, n\}$

1 , Subset Size $N$ **Result:** The Sampled Sub-Dataset of Different Diversity
$$D = \{\mathcal{D}_d^{r_i} | i = 1, \ldots, n\}$$

2 Initialize D = list();

3 **foreach** $i$ *in* $\{1, \ldots, n\}$ **do**

4      Initialize Empty $\mathcal{D}_d^{r_i} \leftarrow \emptyset$;

5      Tag Set $\mathcal{T}_d \leftarrow \emptyset$;

6      **foreach** *Query* $q \in \mathcal{D}$ **do**

7          **if** *Query Tags* $\mathcal{T}_q : |\mathcal{T}_d \cup \mathcal{T}_q| > |\mathcal{T}_d|$ **then**

8              $\mathcal{D}_d^{r_i} \leftarrow \mathcal{D}_d^{r_i} \cup \{q\}$;

9              $\mathcal{T}_d \leftarrow \mathcal{T}_d \cup \mathcal{T}_q$;

10              $\mathcal{D} \leftarrow \mathcal{D} \setminus \{q\}$;

11              **if** $|\mathcal{T}_d|/|\mathcal{T}| > r_i$ **then**

12                  Break;

13      **while** $|\mathcal{D}_d^{r_i}| < N$ **do**

14          **foreach** *Query* $q \in \mathcal{D}$ **do**

15              **if** *Query Tags* $\mathcal{T}_q : |\mathcal{T}_d \cup \mathcal{T}_q| = |\mathcal{T}_d|$ **then**

16                  $\mathcal{D}_d^{r_i} \leftarrow \mathcal{D}_d^{r_i} \cup \{q\}$;

17                  $\mathcal{D} \leftarrow \mathcal{D} \setminus \{q\}$;

18                  **if** $|\mathcal{D}_d^{r_i}| = N$ **then**

19                      Break;

20      D $\leftarrow$ D appends $\mathcal{D}_d^{r_i}$;

21 **return** D

---

## J   CASE FROM INSTAG

We present a sample case in Tab. 10 to illustrate the behavior of INSTAG.

| Metric | GPT-4 Annotation | | Human Annotation (1%) | |
|---|---|---|---|---|
| | Original | Counterfactual | Original | Counterfactual |
| **Tag Precision** | 96.1 | 6.1 | 100 | 0 |
| **Tag Consistency** | 86.6 | 7.8 | 100 | 14.3 |

Table 9: Evaluation for the tagging quality of INSTAG. We design two metrics, tagging precision and consistency, for evaluating INSTAG. We employ GPT-4 to label 4,000 tagging results. And we also employ three human annotators to annotate 1% cases and report their majority voting. We also create counterfactual cases to probe the judgment abilities of different annotators.

| Sample Case | |
|---|---|
| Utterance | |
| Human: | List 3 breakfast foods that are healthy. |
| LLM: | 1.Oatmeal - it's high in fiber, protein, and antioxidants, and can lower cholesterol levels. |
| | 2.Greek yogurt - it's packed with protein, calcium, and probiotics, which support gut health. |
| | 3.Eggs - they're rich in protein, vitamins D and B12, and minerals like selenium, and can help you feel full longer. |
| INSTAG Results | |
| **Tag** | **Explanation** |
| food recommendation | The instruction asks for a list of breakfast foods that are healthy. |
| quantity | The instruction specifies that the list should include three breakfast foods. |
| health | The instruction emphasizes the need for the breakfast foods to be healthy. |

Table 10: One case from SFT data of WizardLM tagged by INSTAG. INSTAG accurately captures the semantics (food, health) and intentions (recommendation, quantity) in the human query.

