# OpenReview forum: "#InsTag: Instruction Tagging for Analyzing Supervised Fine-tuning of Large Language Models"
_ICLR.cc/2024/Conference — ICLR 2024 poster_

### Official Review · Reviewer_NPyy · 2023-10-29

**Soundness:** 3 good
**Presentation:** 4 excellent
**Contribution:** 3 good
**Rating:** 8
**Confidence:** 4

**Summary:**

The authors propose a method for automatically tagging instruction data with labels based on query intent, using ChatGPT to generate queries and applying various rules to normalize and aggregate related tags together. The authors analyse existing datasets using the tagging method, finding that a higher average number of tags, and higher diversity correlate with good performance, matching prior work examining instruction datasets. They then use the tags as a data selection method (maximising complexity and diversity based on the tags), and find they can train a strong model with as few as 6,000 examples. Finally, they ablate their data selection method and find that increasing diversity and complexity improves performance, while increasing dataset size only improves performance up to a point (around 6,000 examples).

**Strengths:**

- The tagging method is interesting, and well-validated both as an analysis and as a selection method. The use of ChatGPT as a tagger, and the rules used to simplify the generated tags, are novel and effective.
- The data selection method appears well-validated, and achieving strong results with only 6,000 examples is impressive. The method outperforms a random selection baseline and other popular open models such as Vicuna and WizardLM.
- The analyses of performance against complexity, diversity, and dataset size are useful, providing useful guidelines for future researchers in data collection and selection.

**Weaknesses:**

- The method’s reliance on a strong tagging model (in this case, ChatGPT) is not explored, despite a very brief mention of training a distilled tagger model at the end of the work. It would be interesting and useful to see how well this method works with openly available models, or over a variety of different quality models (e.g. how does using GPT-4 as a tagger compare? How about Vicuna? etc.).
- The tag-based analysis is somewhat restrictive, as discussed in section 4.3. It would be interesting to take the semantics of the tags themselves into account somehow, since (a) some tags may be closer to each other and so overlap in terms of diversity, and (b) certain tags may express queries that are naturally more complex than others (e.g. ‘solve’ vs ‘inquiry’ tags).
- MT-bench evaluation involves a relatively small number of questions (80), which may be easier to cover with 6,000 examples. I wonder how well the TagLM model would perform with more questions (e.g. the alpacaEval setting, which has 800 examples), or on more traditional benchmarks such as MMLU, Big Bench, etc. In general, it would be interesting to see if the selection method is biased towards certain capabilities compared to others.

Overall, I think this work is strong, and my main qualm is that the evaluation of the data selection method is somewhat weak, only exploring one evaluation setting. However, the method proposed is interesting and novel, and produces useful insights for data selection in instruction tuning.

**Questions:**

- Does the average number of tags per instance correlate well with human intuitions of complexity?
- How well does your distilled tagger perform if used as a tagger for data selection? Did you test with different models?
- How does TagLM perform over other benchmarks (e.g. MMLU, HumanEval)? Are there any particular capabilities or skills it seems to underperform in compared to baselines?

---

> ### Author Response · Authors · 2023-11-13
>
> We appreciate reviewer NPyy for valuable insights and comments. We first present a summary for our update corresponding to the review comments:
>
> **General Response**
>
> Most reviewers expect a better evaluation of TagLM and recommend benchmarks. We fully agree with the reviewers that more comprehensive will strengthen the effectiveness of TagLM. So, we add evaluations on AlpacaEval and present them in Section 4.2 Results and Table 3 in this revision.
>
> As shown in the results, TagLM based on LLaMa-1 and trained with the same 6K samples still outperforms most baselines with the same base models on AlpacaEval such as Alpaca, WizardLM, and Vicuna-13b-v1.1, which further enhances the effectiveness of TagLM and the mechanism of diversity and complexity defined by tags behind the models. But we also notice it falls behind Vicuna-13b-v1.3, which achieves 82.1% win rate on AlpacaEval. TagLM based on LLaMa-2 also has comparable performance with Llama-2-13b-chat on AlpacaEval.
>
> We would love to add human evaluation of TagLM against WizardLM and Vicuna-v1.1 as our 6K samples are mainly collected from wizard and sharegpt datasets. However, such an evaluation is challenging to finish in a week. So, we will add it to the camera-ready version to provide a more objective evaluation.
>
> **Responses to reviewer NPyy’s concerns about the weaknesses and questions**
>
> - W1: We thank reviewer NPyy for proposing this valuable analysis. We ran a quick analysis and found 92% of the 6K SFT samples we used to train TagLM (selected based on ChatGPT tags) is still collected based on local tagger’s tags via the complexity first diverse sampling. This overlap and the F1 on the validation set we reported in the manuscript provide evidence of the consistency between ChatGPT and local tagging. We will train a local tagger-based TagLM and report the end-to-end performance in our camera-ready version. We will release the ChatGPT-tagged data pool and local tagger to promote the development of more precise, cheap, and open-source tag annotators.
>
> - W2: We agree with the two limitations pointed out by reviewer NPyy, especially the second limitation. We have constrained the minimum semantic distance between tags in the normalization described in Section 3 to avoid limitation (a) due to synonym tags. We consider involving the semantic distance between tags in the diversity metric a valuable future work. Limitation (b) is mainly due to the simplicity of our complexity definition.
>
> - W3: Please kindly refer to the general response.
>
> - Q1: Yes, we will add a case study to present the correlation between human intuitions of complexity and our metrics. We notice queries with more tags contain multiple intentions or more complex constraints.
>
> - Q2: Please kindly refer to the response of W1.
>
> - Q3: Please kindly refer to the general response. As for the second question, we have some insights from detailed MT-Bench results in Table 8. We notice TagLM only shows marginal improvements on mathematic questions, which is somehow reasonable as existing works show that enhancing reasoning abilities may require tremendous supervision.

---

> > ### Comment · Reviewer_NPyy · 2023-11-18
> > **Re: comment by authors**
> >
> > Hi, thanks for the response! I can't see the general comment (is it submitted, or maybe the viewers were selected incorrectly?).
> >
> > Your quick analysis is encouraging, suggesting the distillation is effective. I still think it would be interesting to see how the tag efficacy changes with the model used (as the field develops higher-quality models, or if you have a constrained setting where it is only possible to use some lower-quality model). Looking forward to seeing the human correlation results in the final paper.
> >
> > I've read through the other reviews and responses carefully and am keeping my score.

---

### Official Review · Reviewer_jYjU · 2023-10-30

**Soundness:** 3 good
**Presentation:** 3 good
**Contribution:** 2 fair
**Rating:** 6
**Confidence:** 4

**Summary:**

This study illustrates how GPT-4's open-ended topic tagging capability can be instrumental in quantitatively assessing the diversity and complexity of various Instruction Fine-Tuning (IFT) datasets.
Specifically, the complexity of an IFT dataset is gauged by the average number of tags per data point, while its diversity is measured by the proportion of unique tags in the dataset relative to the total number of tags recognized by GPT-4.
Utilizing these novel quantitative metrics, the researchers compared distinct IFT datasets to discern their unique features.
The research further suggests that employing these metrics to filter data can enhance the effectiveness of instruction fine-tuning.
Experimental results in the paper demonstrate the importance of enhancing the diversity and complexity of data points when tuning language models to align with human instructions.

**Strengths:**

- The proposed method is straightforward, intuitive, and simple to implement.
- The study demonstrates that metrics driven by the statistical analysis of automatically generated tags—specifically complexity and diversity—can effectively probe the characteristics of IFT datasets.
- This work provides further empirical evidence that the quality, rather than the quantity, of IFT datasets is crucial for aligning language models successfully.

**Weaknesses:**

- The primary mechanism of the proposed method is dependent on the automated, open-ended tagging capability of GPT-4; therefore, there is a risk that the analysis in this paper might be influenced by any inherent biases present within GPT-4.
- Further examination of data instances categorized by the proposed metrics would be advantageous. Specifically, exploring the semantic or syntactic traits defining IFT datasets identified as diverse and complex by these metrics would be informative.
- The presented work is largely empirical, which may raise concerns within the community regarding the foundational grounding of its results.

**Questions:**

- The proposed method appears to be sufficiently versatile to extend beyond instruction fine-tuning applications. Could you provide additional examples where the efficacy of this method could be demonstrated?

---

> ### Author Response · Authors · 2023-11-13
>
> We appreciate reviewer jYjU's insightful review and valuable suggestions for further investigating our metrics. We first present a summary of our update corresponding to the review comments:
>
> **General Response**
>
> Most reviewers expect a better evaluation of TagLM and recommend benchmarks. We fully agree with the reviewers that more comprehensive will strengthen the effectiveness of TagLM. So, we add evaluations on AlpacaEval and present them in Section 4.2 Results and Table 3 in this revision.
>
> As shown in the results, TagLM based on LLaMa-1 and trained with the same 6K samples still outperforms most baselines with the same base models on AlpacaEval such as Alpaca, WizardLM, and Vicuna-13b-v1.1, which further enhances the effectiveness of TagLM and the mechanism of diversity and complexity defined by tags behind the models. But we also notice it falls behind Vicuna-13b-v1.3, which achieves 82.1% win rate on AlpacaEval. TagLM based on LLaMa-2 also has comparable performance with Llama-2-13b-chat on AlpacaEval.
>
> We would love to add human evaluation of TagLM against WizardLM and Vicuna-v1.1 as our 6K samples are mainly collected from wizard and sharegpt datasets. However, such an evaluation is challenging to finish in a week. So, we will add it to the camera-ready version to provide a more objective evaluation.
>
> **Response to reviewer jYjU’s concerns about the weaknesses and questions**
>
> - W1: We fully agree with the conjecture that ChatGPT may contain a certain inherent bias that influences the tagging results. We also consider it one of our main challenges and develop a normalization procedure to mitigate this issue. For example, ChatGPT sometimes provides associated tags, so we conduct association analysis to combine taggers that always appear together. Besides, this concern is more related to ChatGPT than the methodology of InsTag, as InsTag can easily adopt stronger and unbiased automatic annotators in the future.
>
> - W2: We sincerely appreciate the reviewer pointing out this idea, and this is a great starting point to further dive deep into our metrics.
>
> - W3: We agree with the reviewer jYjU that this work mainly empirically shows the effectiveness of diversity and complexity of instructions. We also consider a more foundational interpretation as our valuable future works.
>
> - Q1: Yes, InsTag can be extended to other scenarios as it generally proposes an automatic tagging method for text. We are tagging intentions and semantics of instructions in this work, but it can also be used in annotating information extraction samples where tags are the expected information, classification tasks where tags are the categories, and so on. However, it is more challenging to tag open-set instructions, and how intention tags reveal the diversity and complexity of the instruction tuning dataset is currently significantly understudied. So, we discuss InsTag in the context of instruction tuning in this manuscript.

---

### Official Review · Reviewer_32G5 · 2023-11-01

**Soundness:** 3 good
**Presentation:** 4 excellent
**Contribution:** 3 good
**Rating:** 5
**Confidence:** 4

**Summary:**

This paper explores the diversity and complexity of supervised data in the alignment process of large language models. It proposes InsTag, an open-set instruction tagging method, which can conveniently evaluate the diversity and complexity of human instructions. Based on this, the authors design a method of selecting human instructions, which can make the model achieve better performance with less supervised training data.

**Strengths:**

1. This paper proposes an automatic method to evaluate the diversity and complexity of human instructions.
2. Based on the proposed InsTag, this paper further presents a method of selecting human instructions, which can potentially reduce the cost of the alignment phase of large language models.

**Weaknesses:**

1. Selecting more diverse and complex samples is an existing idea, and this article is more similar to the implementation and application of this idea.
2. The definition of complexity seems a little strange. Is there any explanation from other papers? If not, I hope the authors can give a more detailed explanation for it.
3. The author is recommended to verify the performance of the proposed method on more benchmarks.

**Questions:**

The description of InsTagger in Section 5 seems too brief. Can you provide more detailed explanations, such as why it was designed and further influences.

---

> ### Author Response · Authors · 2023-11-13
>
> We sincerely appreciate reviewer 32G5 for your comments about motivations and definitions. We first present a summary for our update corresponding to the review comments:
>
> **General Response**
>
> Most reviewers expect a better evaluation of TagLM and recommend benchmarks. We fully agree with the reviewers that more comprehensive will strengthen the effectiveness of TagLM. So, we add evaluations on AlpacaEval and present them in Section 4.2 Results and Table 3 in this revision.
>
> As shown in the results, TagLM based on LLaMa-1 and trained with the same 6K samples still outperforms most baselines with the same base models on AlpacaEval such as Alpaca, WizardLM, and Vicuna-13b-v1.1, which further enhances the effectiveness of TagLM and the mechanism of diversity and complexity defined by tags behind the models. But we also notice it falls behind Vicuna-13b-v1.3, which achieves 82.1% win rate on AlpacaEval. TagLM based on LLaMa-2 also has comparable performance with Llama-2-13b-chat on AlpacaEval.
>
> We would love to add human evaluation of TagLM against WizardLM and Vicuna-v1.1 as our 6K samples are mainly collected from wizard and sharegpt datasets. However, such an evaluation is challenging to finish in a week. So, we will add it to the camera-ready version to provide a more objective evaluation.
>
> **Response to reviewer 32G5’s concerns in the weaknesses**
>
> - W1: We agree with reviewer 32G5 that “diverse and complex samples will benefit instruction tuning” is a widely accepted insight nowadays as methods for diversifying and complexing SFT data emerge. However, we would like to emphasize that the motivation of InsTag is not only to provide solid empirical evidence for this claim but also to investigate reasonable and easy-to-use metrics for diversity and complexity. For example, we try to answer a research question such as both WizardLM and shareGPT may contain “diverse and complex” instructions but which is more “diverse and complex” and how it leads to the final performance. InsTag provides straightforward metrics using open-set tags to quantify these observations, and we show the correlations between our metrics and final performance.
>
> - W2: We thank the reviewer for pointing out the vague of our definition of instruction complexity based on tags. Tags initially inspire this definition in the recommendation system. More tags imply the query is multi-facet, in other words, complex in instruction tuning as it reveals more intentions and constraints. We adopt it for its simplicity and effectiveness. Besides, the analysis in Section 3 and Section 4.3 also shows its effectiveness as a trustworthy metric.
>
> - W3: Please kindly refer to the general response.

---

> > ### Comment · Reviewer_32G5 · 2023-11-23
> >
> > Thanks for your responses!

---

### Official Review · Reviewer_jmr3 · 2023-11-02

**Soundness:** 3 good
**Presentation:** 4 excellent
**Contribution:** 4 excellent
**Rating:** 6
**Confidence:** 5

**Summary:**

This paper presents InsTag, a method to quantify and select/prune sft dataset from a large dataset pool. The authors propose to utilize ChatGPT to tag the intent/topic of the data samples, and use the number of tags to represent complexity and tag coverage for diversity. These metrics are simple from both concept and practice perspectives. The authors give interesting analysis of existing datasets by adopting these two metrics. In the experiments, InsTag automatically selects 6K sft examples and the resulting model achieves comparable performance to WizardLM and Vicuna that are trained on 10x more data examples.

**Strengths:**

1. The proposed data measurement and data selection methods are simple, both concept-wise and practice-wise.
2. The experimental results are strong – TagLM is the only model that achieves such high performance on MT-bench with <10K data examples as far as I know.
3. The proposed approach can be utilized to measure existing datasets, as shown in Figure 2 which are interesting.
4. Table 4 indicates that more sft data does not necessarily give better performance and InsTag is able to select the effective ones.

**Weaknesses:**

1. Evaluation is a bit weak, MT-Bench scores are the only metric used across entire paper – other datasets such as AlpacaEval and human evaluation could further strengthen the claims in the paper.
2. While I appreciate that the authors distill a local tagger in Section 5, it is unknown how much SFT performance would be sacrificed by using the tags from this local tagger. The strong results on Table 3 is from ChatGPT tagger if I understand correctly, and it may be too expensive to use ChatGPT in practice when we have a large data pool to measure.

**Questions:**

NA

---

> ### Author Response · Authors · 2023-11-13
>
> We sincerely appreciate reviewer jmr3 for your comments and insightful suggestions, especially the discussion about our local tagger. We first present a summary for our update corresponding to the review comments:
>
> **General Response**
>
> Most reviewers expect a better evaluation of TagLM and recommend benchmarks. We fully agree with the reviewers that more comprehensive will strengthen the effectiveness of TagLM. So, we add evaluations on AlpacaEval and present them in Section 4.2 Results and Table 3 in this revision.
>
> As shown in the results, TagLM based on LLaMa-1 and trained with the same 6K samples still outperforms most baselines with the same base models on AlpacaEval such as Alpaca, WizardLM, and Vicuna-13b-v1.1, which further enhances the effectiveness of TagLM and the mechanism of diversity and complexity defined by tags behind the models. But we also notice it falls behind Vicuna-13b-v1.3, which achieves 82.1% win rate on AlpacaEval. TagLM based on LLaMa-2 also has comparable performance with Llama-2-13b-chat on AlpacaEval.
>
> We would love to add human evaluation of TagLM against WizardLM and Vicuna-v1.1 as our 6K samples are mainly collected from wizard and sharegpt datasets. However such an evaluation is challenging to finish in a week. So, we will add it to the camera-ready version to provide a more objective evaluation.
>
> **Responses to reviewer jmr3’s concerns about the weaknesses**
>
> - W1: We thank reviewer jmr3 for recommending AlpacaEval and human evaluation. We have added more evaluation to strengthen our claims. Please kindly refer to the General Response.
>
> - W2: Yes, the result in Table 3 is from the ChatGPT tagger and our corresponding normalization procedure. And we also agree with the reviewer that annotating tags with ChatGPT is very expensive. To address reviewer jmr3’s concern, we ran a quick analysis and found 92% of the 6K SFT samples we used to train TagLM (selected based on ChatGPT tags) are still collected based on local tagger’s tags via the complexity first diverse sampling. This overlap and the F1 on the validation set we reported in the manuscript provide evidence of the consistency between ChatGPT and local tagging. We will train a local tagger-based TagLM and report the end-to-end performance in our camera-ready version. We will release the ChatGPT-tagged data pool and local tagger to promote the development of more precise and cheap tag annotators.

---

> > ### Comment · Reviewer_jmr3 · 2023-11-22
> > **Thank you for the response**
> >
> > Thank you for the response and adding the AlpacaEval results.

---

### Meta-Review · Area_Chair_Arm3 · 2023-12-17

**Metareview:**

The paper introduces InsTag, an open-set instruction tagging method, to quantify and analyze the diversity and complexity of human instructions for LLMs during SFT. By generating fine-grained tags, the study identifies that aligning language models benefits from diverse and complex instructions. InsTag is then used to select effective samples for SFT, resulting in TagLM models outperforming larger datasets on MT-Bench evaluation. The findings show the importance of instruction diversity and complexity. Reviewers highlighted several strengths, including novelty, simple metrics, good experiments, useful analysis, and helpful guidelines. However, there are also various limitations. Some suggestions include providing a more detailed explanation to clarify the concept of the definition of complexity and examining performance on more benchmarks, among others. It's positive that the authors have generally agreed to address these aspects in the revised version.

**Justification For Why Not Higher Score:**

The work has several limitations; it does not stand out among the papers I have reviewed.

**Justification For Why Not Lower Score:**

Good work; I don't see a reason to reject this work, despite its several limitations.

---

### Decision · Program_Chairs · 2024-01-16

Accept (poster)